# Tobacco smoking and smokeless tobacco use among people living with HIV in Zambia: Findings from a 2023 National NCD/HIV Survey

Cosmas Zyambo[1]*, Paul Somwe[2], Chomba Mandyata[1], Mwiche Musukuma[3], Phoebe Bwembya[1], Henry Phiri[4], Malizgani P. Chavula[1], Hikabasa Halwindi[1], Joseph Zulu[5], Wilbroad Mutale[5]

1 Department of Community and Family Medicine, School of Public Health, University of Zambia, 2 Centre for Infectious Disease Research in Zambia, 3 Department of Epidemiology and Biostatistics, School of Public Health, University of Zambia, 4 |Ministry of Health in Zambia, Lusaka, Zambia, 5 Department of Health Policy, School of Public Health, University of Zambia

* czyambo256@gmail.com

## Abstract

### Background

People living with HIV (PLWH) who use tobacco face significant public health risks compared to non-users, including an average loss of 12.3 years of life expectancy. Tobacco use increases the likelihood of non-communicable diseases (NCDs), such as cardiovascular diseases, hypertension, diabetes mellitus, and non-AIDS-related cancers.

### Aim

This study investigated factors associated with tobacco smoking and smokeless tobacco (SLT) use among PLWH in Zambia.

### Methods

Data were obtained from a national cross-sectional survey involving 5,204 PLWH from 193 clinics across Zambia's 10 provinces. Tobacco smoking, SLT use, behavioral patterns, and clinical characteristics were assessed. Logistic regression was used to determine unadjusted (UOR) and adjusted odds ratios (AOR) at a 95% confidence interval (CI).

### Results

Among the 5,204 PLWH surveyed, 9.7% were current tobacco smokers (21.9% men, 3.7% women), while 1.4% used smokeless tobacco (1.81% men, 1.26% women). In the multivariable analysis, several factors were identified as predictors of tobacco smoking. Male individuals had significantly higher odds of smoking

**Data availability statement:** Data cannot be shared publicly because of analysis which is currently on going for other manuscripts. Data will be available from the National health Ethics authority website (https://www.nhra.org.zm/) for researchers who meet the criteria for access to confidential data. For further details regarding the dataset, kindly reach out to the National health research ethics board chairperson , Prof. Patrick Musonda. pmuzho@hotmail.com.

**Funding:** Global Fund. The funders had no role in study design, data collection and analysis, decision to publish, or preparation of the manuscript.

**Competing interests:** The authors have declared that no competing interests exist.

(AOR: 4.81, 95% CI: 3.36–6.90). In contrast, higher educational attainment was associated with lower odds of smoking (AOR: 0.29, 95% CI: 0.16–0.52). Alcohol consumption was associated with an increased likelihood of smoking (AOR: 4.97, 95% CI: 2.93–8.44). Additionally, overweight or obese individuals were less likely to smoke, with adjusted odds ratios of 0.55 (95% CI: 0.35–0.85) and 0.36 (95% CI: 0.17–0.79), respectively. Non-adherence to antiretroviral therapy (ART) was also associated with higher smoking rates (AOR: 1.75, 95% CI: 1.14–2.67). Similarly, several factors were identified as predictors of smokeless tobacco (SLT) use. Individuals with an annual income exceeding 4,000 ZMW had lower odds of using SLT (AOR: 0.31, 95% CI: 0.14–0.73). In contrast, alcohol users exhibited significantly higher odds of SLT use (AOR: 14.74, 95% CI: 1.99–109.02). Furthermore, non-adherence to ART was associated with an increased likelihood of SLT use (AOR: 3.32, 95% CI: 1.54–7.17).

## Conclusions

Our findings highlight the urgent need for targeted interventions to reduce tobacco use among PLWH in Zambia. Integrating these measures within the existing healthcare framework can maximize impact. Gender-specific programs addressing unique risk factors, alongside economic empowerment initiatives for low-income females, could help curb SLT use. Additionally, reinforcing ART adherence through tobacco cessation counseling within HIV care settings may lower smoking rates. Given the strong association between alcohol consumption and tobacco use, structured behavioral interventions and support programs should also be prioritized. Strengthening collaborations between health authorities and community organizations can further enhance accessibility and outreach. By embedding these strategies within primary care and ART clinics, Zambia can effectively reduce tobacco use among PLWH, ultimately improving overall health outcomes and strengthening HIV management efforts.

## Background

The introduction and increased accessibility of combination antiviral therapy (ART) have significantly reduced AIDS-related morbidity and mortality among people living with HIV (PLWH) [1]. However, HIV still claims nearly 1 million lives annually, with the majority of these deaths occurring in sub-Saharan Africa (SSA) [2,3]. While the estimated general population smoke, the prevalence of smoking in PLWH is 2–3 times higher than the general population [4–7], thus posing a significant public health concerns which includes loss of an average 12.3 life-years more as compared to PLWH non-smokers [7]. PLWH who smoke face a higher risk of non-communicable diseases (NCDs) such as cardiovascular diseases, hypertension, diabetes mellitus, and non-AIDS-related cancers [8]. They are also more likely to be non-adherent to ART, have unsuppressed HIV viral loads, and lower CD4

counts compared to non-smoking PLWH [9–11]. Crothers et al. conducted a study on US veterans, finding mortality rates per 100 person-years to be 1.76 for HIV-negative non-smokers, 2.45 for HIV-positive non-smokers, and 5.48 for HIV-positive current smokers [12].

Zambia, a lower-middle-income country (LMIC) has one of the highest numbers of PLWH in the sub-Saharan Africa (SSA) region with more than 11% of the adults between 15–49 living with HIV [13]. The prevalence of tobacco smoking has been varying depending on data survey being used. According to the Zambia Demographic Health Surveys (DHS), the overall prevalence of tobacco smokers ranges from 9.9% to 12.0% [14], and 1% among women and 19.6% among men [15]. Global Youth Tobacco Survey (GYTS), which monitors tobacco smoking among school-going adolescents, reports an overall prevalence ranging from 6.8% to 10.5% [16] and the STEPS survey by the WHO in Zambia reveals a tobacco use prevalence range of 8.7% to 12.3% [17]. In all these studies, the rural vs urban and male vs female differentials were significantly observed.

Consistent with findings from other regions, HIV has been associated with increased likelihood of tobacco use in sub-Saharan Africa (SSA), even after adjusting for demographic, socioeconomic, and sexual risk factors. Mdege et al [4] analyzed data from DHS across 28 LMICs, including Zambia, and found that both men and women living with HIV had higher rates of tobacco use including, tobacco smoking, and SLT (also known as Insunko in Zambia) compared to HIV negative counterparts. Similarly, Murphy et al [18] conducted a subsequent analysis across 25 SSA countries, reaffirming that people living with HIV are more likely to use tobacco and smokeless tobacco than their HIV-negative counterparts. More recently study by Kress et al [19]., utilized data from the 2018 Zambia DHS, and reported a higher prevalence of tobacco use among both women and men living with HIV compared to those without HIV. These studies primarily rely on household surveys, which may not capture patterns of tobacco use among PLWH receiving health care services. This underscores the need for facility-based studies to provide more comprehensive insights into tobacco use behaviors, clinical correlates, and targeted intervention opportunities with HIV care settings.

To our knowledge, there is a lack of in-depth analyses examining the prevalence of tobacco smoking and SLT use, along with their associated factors among PLWH receiving regular care in HIV health facilities. This study aims to bridge that gap by estimating the prevalence of tobacco smoking and SLT use while identifying key predictors influencing these behaviors. Notably, Zambia currently lacks a structured smoking cessation policy for individuals attending routine HIV clinics. A structured smoking cessation policy could significantly improve health outcomes among PLWH by integrating tobacco cessation support into routine HIV care. Given the strong association between smoking and non-adherence to ART, such a policy could enhance treatment effectiveness, reduce complications associated with tobacco use, and improve overall patient well-being. Furthermore, implementing tobacco cessation interventions within HIV clinics would allow for targeted counselling, pharmacological support, and behavioral therapies, ensuring comprehensive care tailored to the needs of PLWH. The study findings would address tobacco use among PLWH and provide valuable insights into key risk factors, such as gender differences, socioeconomic influences, and alcohol consumption. These results could guide policymakers in designing a tailored cessation framework, incorporating evidence-based strategies to reduce tobacco use. By leveraging Zambia's existing healthcare infrastructure, particularly ART clinics and primary care settings, a structured policy could enhance accessibility and engagement in cessation programs, ultimately reducing the burden of NCDs and improving overall health outcomes among PLWH.

## Methodology

### Study design and setting

This was a nationally cross-sectional representative survey on non-communicable diseases among PLWH in the 10 provinces of Zambia (Fig 1: Provinces and districts of the study area in Zambia). The survey included HIV+ clients on ART from the health facilities that provide ART services country-wide. Health facilities offering HIV treatment and management services and use the national electronic health record system known as SMARTCARE were included. All HIV+ adults

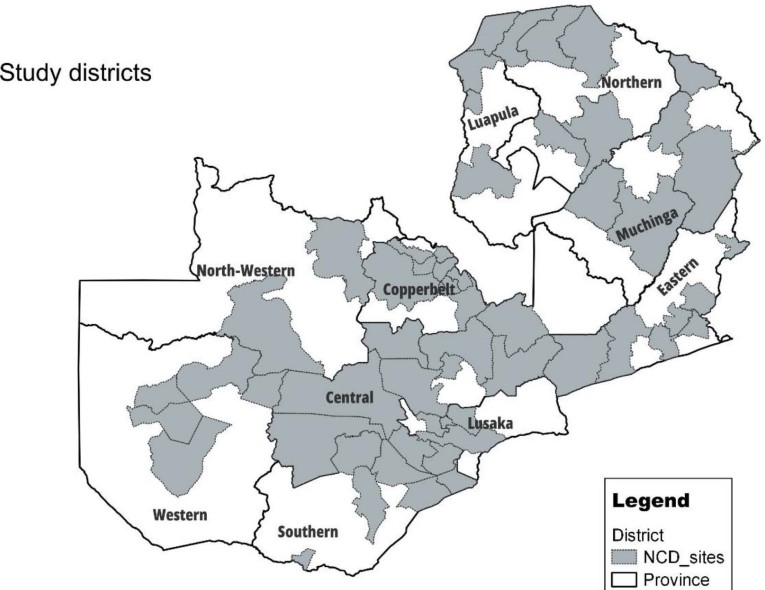

Study districts

**Fig 1. Provinces and districts of the study area in Zambia.** The map was created using QGIS 3.28.13 and base map data used was downloaded from Zambia - Subnational Administrative Boundaries - Humanitarian Data Exchange (humdata.org). The Humanitarian data exchange has the "Creative Commons Attribution International (CC BY)" license. This license permits us to share and adapt maps with necessary attributes. Please see link to the licence: Data Licenses - Humanitarian Data Exchange (humdata.org).

(aged 18 years and above) on ART at the time of the survey and were accessing ART services in the sampled health facility that offers ART services were eligible for enrolment.

### Sample size calculation and enrolment of participants

To estimate the sample size, we used the prevalence formula assuming an infinite population. With no reliable estimate of the prevalence of NCDs among PLWH, we used a prevalence of 50% to estimate the sample size. We estimated a sample size of 385 with a two-sided 95% confidence interval and a 10% margin of error. To ensure adequate representation across the different age-groups (18–29, 30–44, 45–59, and 60 + years), by rural-urban and by sex, we multiplied the calculated sample size by the total number of domains, i.e., 8. This yielded a sample of 3,080. To address the issue of clustering and non-response, we multiplied the sample size by a design effect of 1.5 and adjusted for a 20% non-response rate, giving us a total sample size of 5775.

We used a two-stage cluster sampling technique to select a nationally representative sample of health facilities providing ART services. A cluster was defined as a health facility (i.e., clinic) made up of HIV+ patients. Our sampling frame consisted a list of all health facilities providing ART services with a TXR of at least 500 from the electronic health record system (SMARTCARE). In the first sampling stage, facilities were stratified by Province, then secondly by rural-urban creating a total of 20 strata. To allocate the total sample size of 5775 across the 20 strata with a fixed sample take of 35 participants from each health facility, we used probability proportional to size to select 165 health facilities from the sampling frame. Those who were missing outcome variables of interest were excluded from the analysis. Data from the participants was collected between 08/07/2023–18/07/2023. To reduce selection bias, four participants were systematically sampled per day, two in the morning and two in the afternoon. Because the main focus of the parent study was NCD's among PLWH, all pregnant women and breastfeeding mothers were excluded from the study. Inclusion of the age group of 18–29 years in this study is crucial because young adults are at a pivotal life stage marked by transitions in education,

employment, and social environments, all of which influence health behaviors, including tobacco use. Peer pressure, growing independence, and exposure to various stressors make them particularly vulnerable to initiating and sustaining tobacco use. From a healthcare perspective, understanding tobacco use patterns in this group enables early intervention, helping to mitigate long-term health risks, especially among PLWH, where tobacco exacerbates complications. Insights from this analysis can inform youth-focused prevention strategies, awareness campaigns, and policy measures designed to curb tobacco use before it becomes deeply ingrained

## Measures and variables

**Outcomes of interest.**  We have two primary outcomes of interest: 1) Tobacco smoking as asked from the questionnaire "Do you currently smoke any tobacco products, such as cigarettes, Shisha, cigars or pipes?" (Yes/ No); 2) Smokeless tobacco (SLT) use as asked from the questionnaire "Do you currently use any smokeless tobacco products such as *snuff, chewing tobacco*?" (Yes/No)

**Independent variables.**

1) **Sociodemographic variables** included sex (male or female) age, education level (No formal schooling, less than primary, Primary school completed, Junior Secondary school completed, Secondary Higher school completed, College/University completed and post graduate degree), Income (< K4000.00 vs ≥ K4000.00); occupation (employed vs non-employed), marital status (single, married or widowed/separated); current smoking status (smoker vs non-smoker)

2) **Behavior variables,** alcohol use as asked "have you consumed any alcohol within the past 30 days?" (Yes or No)

3) **Clinical variables** included diabetes, history of hypertension, history of metabolic diseases (diabetes and dyslipidemia), history of cardiovascular diseases (heart attack, stroke), the question was on each disease "have you ever been told by a doctor or other health worker that you have raised blood pressure, raised blood sugar or diabetes? Raised cholesterol levels? heart attack or stroke? the answer was Yes or No.

## Statistical analysis

The characteristics of tobacco smoking and SLT use were analyzed for the overall study population. Continuous variables were summarized using means and Standard Deviations (SD), while categorical variables were presented as frequencies with percentages. To assess associations between exposure variables and tobacco use outcomes, both univariate and multivariable logistic regression models applied. Odds ratios (ORs) and the corresponding 95% confidence intervals (CIs) were calculated to estimate the strength of associations, with adjusted ORs (AORs) accounting for potential confounders. Clinically relevant variables were predetermined and included in the final model regardless of the univariate statistical significance (age, sex, clinical variables) with additional variables included based on a univariate statistical significance ($p < 0.05$) with tobacco smoking and SLT status. Statistical significance level was set at $p < 0.05$ (two-tailed test). Analysis was performed using StataCorp. 2017. Stata Statistical Software: Release 15. College Station, TX: StataCorp LLC.

## Ethical consideration

Ethics approval was obtained from UNZABREC (REF. 3007−2022). The protocol was subsequently submitted to the Zambia National Health Research Authority (Ref No: NHRA0000011/12/09/2022) for registration and approval. Permission was sought from the Ministry of Health to allow data to be collected from the selected health facilities and for their staff to participate in the study.

Informed consent was secured from all participants before their involvement in the study. We ensured the confidentiality of the data obtained. Each participant provided written informed consent, facilitated by a trained member of the study team. Potential respondents had the opportunity to ask any questions about the research before deciding whether to

participate. Those who agreed to participate were interviewed. Participants received written informed consent forms and information sheets detailing the study, its risks and benefits, and emphasizing the protection of confidentiality.

## Results

### Population and distribution

This study involved 5,204 PLWH who participated in the survey, out of which 3,495 (67.2%) were females and 1709 (32.8%) were males with a median age of 43 (IQR: 34, 51) years. Educational attainment varied, with 2,331 (44.8%) having completed secondary education and 468 (9.0%) having attained higher education levels (i.e., college or university). Most participants were either currently married or cohabiting, accounting for 2,929 (56.3%) respondents, while the remaining participants had varying marital status. Employment status revealed that 1,041 (20.0%) were formally employed, 1,976 (38.0%) were unemployed, and 2079 (40.0%) were in informal employment. The average yearly income of the participants was categorized into two groups: those earning ≤4,000 ZMW and those earning >4,000 ZMW. 2,389 (45.9%) of the participants fell into the lower income category, while 1,210 (23.3%) were in the higher income category, and 1,605 (30.8%) had missing income data. The study also considered the location of the participants, with 1,627 (31.3%) residing in rural areas and 3,577 (68.7%) in urban areas. Of the 5,204 PLWH, 1756 (33.7%) were either overweight or obese, 872 (18.3%) had a history of Hypertension, 136 (6.5%) raised blood sugar or diabetes, 5 (10.4%) raised cholesterol, 85 (1.6%) heart attack/ stroke and 371(7.1%) were non-adherent to ART (Table 1).

### Prevalence and pattern of tobacco smoking

The overall prevalence of tobacco smoking among PLWH was 503 (9.7%), with a significantly higher proportion of men 375 (74.6%) compared to females 128 (25.4%). The highest smoking prevalence was observed among individuals aged 30–44 (48.9%), those residing in urban areas 335 (66.6%), and individuals with no formal education or only primary education (51.8%). Smoking was also more common among those with Informal (40%), individuals who were currently married or Cohabitating 314 (62.4%), and those with average annual income of ≤4,000 ZMW 256 (51%). Among current smokers 142 (28.2%) were overweight or obese, while 376 (74.8%) reported alcohol consumption in the past 12 months. Regarding clinical conditions, 14.7% of smokers had high blood pressure, 1.2% had a history of a heart attack or stroke, and 1.2% had raised blood sugar levels. Additionally, nearly 16% of tobacco smokers were non-adherent to ART (Table 1). Among tobacco users, manufactured cigarettes were the most commonly used product, with 255(44.7%) smoking daily and 102 (20.3%) smoking weekly, with a higher prevalence among men. Shisha use was more common on a weekly basis (36%) compared to daily use (29%), with 39.7% of men (n = 149) smoking shisha weekly compared to a lower proportion of women (Table 2).

### Prevalence and pattern of smokeless tobacco

The overall prevalence of smokeless tobacco among PLWH was 1.4%, with a higher proportion of women (58.7%) using smokeless tobacco compared to men. The highest prevalence was observed among individuals aged 30–44 (53.3%), those residing in urban areas 57 (76%), and individuals with no formal education or only primary education (57.4%). Smokeless tobacco was also more common among unemployed individuals (39%), those who were married or Cohabitating 40 (53.3%), and those with an annual income of ≤4,000 ZMW 45 (60%). Among current SLT users, 19 (25.3%) were overweight or obese. while 60 (80%) reported consuming alcohol in the last 12 months. Regarding clinical conditions, 24% had high blood pressure (24%), 2.7% had a history of heart attack or stroke, 6.7% had raised blood sugar levels. Additionally, more than 17% of SLT users were non-adherent to ART (Table 1).

Women were more likely than men to use SLT daily, with the most common methods of consumption being snuffing by mouth (43.2%), snuffing by nose (31.8%), and chewing tobacco (31.8%) (Table 2)

**Table 1. Proportions of sociodemographic, behavioural, and clinical variables (tobacco smoking and smokeless tobacco).**

| Characteristics | Overall | Smoke tobacco | | Smokeless tobacco | |
|---|---|---|---|---|---|
| | N (% of total) | No n (%) | Yes n (%) | No n (%) | Yes n (%) |
| | 5204 (100) | 4701 (90.3) | 503 (9.7) | 5129 (98.6) | 75 (1.4) |
| **Gender** | | | | | |
| Female | 3495 (67.2) | 3367 (71.6) | 128 (25.4) | 3451 (67.3) | 44 (58.7) |
| Male | 1709 (32.8) | 1334 (28.4) | 375 (74.6) | 1678 (32.7) | 31 (41.3) |
| **Age (years)** | | | | | |
| median (IQR) | | | | | |
| 18-29 | 724 (13.9) | 676 (14.4) | 48 (9.5) | 718 (14.0) | 6 (8.0) |
| 30-44 | 2202 (42.3) | 1956 (41.6) | 246 (48.9) | 2162 (42.2) | 40 (53.3) |
| 45-59 | 1836 (35.3) | 1661 (35.3) | 175 (34.8) | 1811 (35.3) | 25 (33.3) |
| 60+ | 422 (8.1) | 391 (8.3) | 31 (6.2) | 419 (8.2) | 3 (4.0) |
| Missing | 20 (0.4) | 17 (0.4) | 3 (0.6) | 19 (0.4) | 1 (1.3) |
| **Education** | | | | | |
| No formal schooling | 1131 (21.7) | 1002 (21.3) | 129 (25.6) | 1108 (21.6) | 23 (30.7) |
| Primary | 1271 (24.4) | 1139 (24.2) | 132 (26.2) | 1251 (24.4) | 20 (26.7) |
| Secondary | 2331 (44.8) | 2118 (45.1) | 213 (42.3) | 2301 (44.9) | 30 (40.0) |
| College/University | 468 (9.0) | 439 (9.3) | 29 (5.8) | 466 (9.1) | 2 (2.7) |
| Missing | 3 (0.1) | 3 (0.1) | 0 (0.0) | 3 (0.1) | 0 (0.0) |
| **Marital status** | | | | | |
| Never married | 705 (13.5) | 639 (13.6) | 66 (13.1) | 696 (13.6) | 9 (12.0) |
| Currently married/Cohabitating | 2929 (56.3) | 2615 (55.6) | 314 (62.4) | 2889 (56.3) | 40 (53.3) |
| Divorced/Separated/Widowed | 1551 (29.8) | 1430 (30.4) | 121 (24.1) | 1525 (29.7) | 26 (34.7) |
| Missing | 19 (0.4) | 17 (0.4) | 2 (0.4) | 19 (0.4) | 0 (0.0) |
| **Employment status** | | | | | |
| Formal employment | 1041 (20.0) | 931 (19.8) | 110 (21.9) | 1023 (19.9) | 18 (24.0) |
| Informal employment | 2079 (40.0) | 1861 (39.6) | 218 (43.3) | 2053 (40.0) | 26 (34.7) |
| Unemployed | 1976 (38.0) | 1808 (38.5) | 168 (33.4) | 1947 (38.0) | 29 (38.7) |
| Missing | 108 (2.1) | 101 (2.1) | 7 (1.4) | 106 (2.1) | 2 (2.7) |
| **Average yearly income** | | | | | |
| ≤4,000 ZMW | 2389 (45.9) | 2133 (45.4) | 256 (50.9) | 2344 (45.7) | 45 (60.0) |
| >4,000 ZMW | 1210 (23.3) | 1101 (23.4) | 109 (21.7) | 1202 (23.4) | 8 (10.7) |
| Missing | 1605 (30.8) | 1467 (31.2) | 138 (27.4) | 1583 (30.9) | 22 (29.3) |
| **Location** | | | | | |
| Rural | 1627 (31.3) | 1459 (31.0) | 168 (33.4) | 1609 (31.4) | 18 (24.0) |
| Urban | 3577 (68.7) | 3242 (69.0) | 335 (66.6) | 3520 (68.6) | 57 (76.0) |
| **Behavior variables** | | | | | |
| **Consumed alcohol within past 12m*** | | | | | |
| No | 494 (24.4) | 471 (10.0) | 23 (4.6) | 492 (9.6) | 2 (2.7) |
| Yes | 1527 (75.6) | 1151 (24.5) | 376 (74.8) | 1467 (28.6) | 60 (80.0) |
| **Clinical variables** | | | | | |
| **BMI** | | | | | |
| Normal weight (18.5–24.9) | 2915 (56.0) | 2578 (54.8) | 337 (67.0) | 2866 (55.9) | 49 (65.3) |
| Underweight (≤18.4) | 453 (8.7) | 374 (8.0) | 79 (15.7) | 444 (8.7) | 9 (12.0) |
| Overweight (25–29.9) | 1166 (22.4) | 1103 (23.5) | 63 (12.5) | 1156 (22.5) | 10 (13.3) |

*(Continued)*

| Characteristics | Overall | Smoke tobacco | | Smokeless tobacco | |
|---|---|---|---|---|---|
| | N (% of total) | No n (%) | Yes n (%) | No n (%) | Yes n (%) |
| | 5204 (100) | 4701 (90.3) | 503 (9.7) | 5129 (98.6) | 75 (1.4) |
| Obese (≥30) | 590 (11.3) | 570 (12.1) | 20 (4.0) | 585 (11.4) | 5 (6.7) |
| Missing | 80 (1.5) | 76 (1.6) | 4 (0.8) | 78 (1.5) | 2 (2.7) |
| **Told by a doctor or other health worker that you have raised BP or HTN*** | | | | | |
| No | 3883 (81.7) | 3507 (74.6) | 376 (74.8) | 3833 (74.7) | 50 (66.7) |
| Yes | 872 (18.3) | 798 (17.0) | 74 (14.7) | 854 (16.7) | 18 (24.0) |
| **Told by a doctor or other health worker that you have raised blood sugar or diabetes*** | | | | | |
| No | 1952 (93.5) | 1775 (37.8) | 177 (35.2) | 1922 (37.5) | 30 (40.0) |
| Yes | 136 (6.5) | 130 (2.8) | 6 (1.2) | 131 (2.6) | 5 (6.7) |
| **Told by a doctor or other health worker that you have raised cholesterol*** | | | | | |
| No | 43 (89.6) | 42 (0.9) | 1 (0.2) | 42 (0.8) | 1 (1.3) |
| Yes | 5 (10.4) | 5 (0.1) | 0 (0.0) | 5 (0.1) | 0 (0.0) |
| **Heart attack or stroke** | | | | | |
| No | 5119 (98.4) | 4622 (98.3) | 497 (98.8) | 5046 (98.4) | 73 (97.3) |
| Yes | 85 (1.6) | 79 (1.7) | 6 (1.2) | 83 (1.6) | 2 (2.7) |
| **ART adherence status** | | | | | |
| Adherent | 4833 (92.9) | 4410 (93.8) | 423 (84.1) | 4771 (93.0) | 62 (82.7) |
| Non-adherent | 371 (7.1) | 291 (6.2) | 80 (15.9) | 358 (7.0) | 13 (17.3) |

## Predictors of tobacco smoking and smokeless tobacco

The multivariable analysis identified the factors associated with tobacco smoking among the PLWH. The adjusted odds ratios (AORs) revealed significant associations between gender, education level, alcohol consumption, BMI, and ART adherence. Males had a significantly higher likelihood of smoking compared to females, with an AOR of 4.81 (95% CI: 3.36–6.90). Higher educational attainment was associated with a lower likelihood of smoking with AORs decreasing as education level increased (primary [AOR 0.81 (0.52–1.26)], secondly [AOR 0.46 (0.30–0.69)], college/university [AOR 0.29 (0.16–0.52)]. Those who consumed alcohol in the past 12 months had increased odds of smoking, with an AOR of 4.97 (2.93–8.44). Overweight or obese participants were less likely to smoke than those with a normal BMI (AORs of 0.55, 95% CI: 0.35–0.85 and AOR 0.36, 95% CI: 0.17–0.79, respectively). Non-adherence to antiretroviral therapy (ART) was associated with higher odds of (AOR of 1.75, 95% CI 1.14–2.67). (Table 3). Additionally, we investigated the factors associated with current SLT. Significant associations were found with income level, alcohol consumption, and ART adherence. Participants with an annual income of ≤K4,000 were more likely to use smokeless tobacco compared to those earning >K4,000 (AOR of 0.31, 95% CI: 0.14–0.73). Those who consumed alcohol in the past 12 months had significantly higher odds of using smokeless tobacco, (AOR 14.74, 95% CI: 1.99–109.02). Non-adherence to ART was associated with a higher likelihood of SLT use (AOR of 3.32, 95% CI: 1.54–7.17) (Table 4).

**Table 2. Patterns of tobacco use among participants, stratified by type (smoke and smokeless).**

| Types of Tobacco | Overall | Sex | |
|---|---|---|---|
| | n<br>(% of total) | Female<br>n (%) | Male<br>n (%) |
| **Smoke Tobacco** | | | |
| **Manufactured cigarettes** | | | |
| Daily | 225 (44.7) | 28 (21.9) | 197 (52.5) |
| Weekly | 102 (20.3) | 27 (21.1) | 75 (20.0) |
| No | 176 (35.0) | 73 (57.0) | 103 (27.5) |
| **Hand rolled cigarettes** | | | |
| Daily | 174 (34.6) | 23 (18.0) | 151 (40.3) |
| Weekly | 153 (30.4) | 32 (25.0) | 121 (32.3) |
| No | 176 (35.0) | 73 (57.0) | 103 (27.5) |
| **Pipes full of tobacco** | | | |
| Daily | 162 (32.2) | 23 (18.0) | 139 (37.1) |
| Weekly | 165 (32.8) | 32 (25.0) | 133 (35.5) |
| No | 176 (35.0) | 73 (57.0) | 103 (27.5) |
| **Cigars, cheroots or cigarillos** | | | |
| Daily | 159 (31.6) | 23 (18.0) | 136 (36.3) |
| Weekly | 168 (33.4) | 32 (25.0) | 136 (36.3) |
| No | 176 (35.0) | 73 (57.0) | 103 (27.5) |
| **Shisha** | | | |
| Daily | 146 (29.0) | 23 (18.0) | 123 (32.8) |
| Weekly | 181 (36.0) | 32 (25.0) | 149 (39.7) |
| No | 176 (35.0) | 73 (57.0) | 103 (27.5) |
| **Total** | **503 (100)** | **128 (25.5)** | **375 (74.5)** |
| **Smokeless tobacco** | | | |
| **Snuff, by mouth** | | | |
| Daily | 26 (34.7) | 19 (43.2) | 7 (22.6) |
| Weekly | 5 (6.7) | 3 (6.8) | 2 (6.5) |
| No | 44 (58.7) | 22 (50.0) | 22 (71.0) |
| **Snuff by nose** | | | |
| Daily | 21 (28.0) | 14 (31.8) | 7 (22.6) |
| Weekly | 10 (13.3) | 8 (18.2) | 2 (6.5) |
| No | 44 (58.7) | 22 (50.0) | 22 (71.0) |
| **Chewing tobacco** | | | |
| Daily | 19 (25.3) | 14 (31.8) | 5 (16.1) |
| Weekly | 12 (16.0) | 8 (18.2) | 4 (12.9) |
| No | 44 (58.7) | 22 (50.0) | 22 (71.0) |
| **Betel, quid with tobacco** | | | |
| Daily | 15 (20.0) | 10 (22.7) | 5 (16.1) |
| Weekly | 16 (21.3) | 12 (27.3) | 4 (12.9) |
| No | 44 (58.7) | 22 (50.0) | 22 (71.0) |
| **Total** | **75 (100)** | **44 (58.7)** | **31 (41.3)** |

**Table 3. Factors associated with tobacco smoking among PLHIV.**

| Characteristics | Crude OR (95% CI) | chi2 Statistic | p-value | Adjusted OR (95% CI) | p-value |
|---|---|---|---|---|---|
| **Gender** | | | | | |
| Female | Ref | 410.00 | <0.001 | Ref | **<0.001** |
| Male | 7.39 (5.99-9.13) | | | 4.81 (3.36-6.9) | |
| **Age (years)** | | | | | |
| 18-29 | Ref | 16.82 | 0.001 | Ref | 0.080 |
| 30-44 | 1.77 (1.28-2.44) | | | 1.48 (0.85-2.57) | |
| 45-59 | 1.48 (1.06-2.07) | | | 0.99 (0.56-1.78) | |
| 60+ | 1.12 (0.70-1.78) | | | 0.91 (0.42-1.98) | |
| **Education** | | | | | |
| No formal schooling | Ref | 12.52 | 0.006 | Ref | **<0.001** |
| Primary | 0.90 (0.70-1.16) | | | 0.81 (0.52-1.26) | |
| Secondary | 0.78 (0.62-0.98) | | | 0.46 (0.30-0.69) | |
| College/University | 0.51 (0.34-0.78) | | | 0.29 (0.16-0.52) | |
| **Marital status** | | | | | |
| Never married | Ref | 10.26 | 0.006 | | |
| Currently married/Cohabitating | 1.16 (0.88-1.54) | | | | |
| Divorced/Separated/Widowed | 0.82 (0.60-1.12) | | | | |
| **Employment status** | | | | | |
| Formal employment | Ref | 5.66 | 0.059 | | |
| Informal employment | 0.99 (0.78-1.26) | | | | |
| Unemployed | 0.79 (0.61-1.01) | | | | |
| **Average yearly income** | | | | | |
| ≤4,000 ZMW | Ref | 2.62 | 0.106 | Ref | 0.110 |
| >4,000 ZMW | 0.82 (0.65-1.04) | | | 0.76 (0.55-1.06) | |
| **Location** | | | | | |
| Rural | Ref | 1.17 | 0.280 | | |
| Urban | 0.90 (0.74-1.09) | | | | |
| **Behavior variables** | | | | | |
| **Consumed alcohol within past 12m** | | | | | |
| No | Ref | 117.51 | <0.001 | Ref | **<0.001** |
| Yes | 6.69 (4.33-10.33) | | | 4.97 (2.93-8.44) | |
| **Clinical variables** | | | | | |
| **BMI** | | | | | |
| Normal weight (18.5–24.9) | Ref | 100.35 | <0.001 | Ref | **0.001** |
| Underweight (≤18.4) | 1.62 (1.24-2.11) | | | 1.45 (0.93-2.26) | |
| Overweight (25–29.9) | 0.44 (0.33-0.58) | | | 0.55 (0.35-0.85) | |
| Obese (≥30) | 0.27 (0.17-0.43) | | | 0.36 (0.17-0.79) | |
| **Told by a doctor or other health worker that you have raised BP or HTN** | | | | | |
| No | Ref | 1.22 | 0.269 | Ref | 0.870 |
| Yes | 0.86 (0.67-1.12) | | | 0.97 (0.64-1.45) | |

*(Continued)*

**Table 3.** (Continued)

| Characteristics | Crude OR (95% CI) | chi2 Statistic | p-value | Adjusted OR (95% CI) | p-value |
|---|---|---|---|---|---|
| **Told by a doctor or other health worker that you have raised blood sugar or diabetes** | | | | | |
| No | Ref | 4.10 | 0.043 | | |
| Yes | 0.46 (0.20-1.06) | | | | |
| **Heart attack or stroke** | | | | | |
| No | Ref | 0.73 | 0.391 | Ref | 0.22 |
| Yes | 0.71 (0.31-1.63) | | | 0.49 (0.15-1.54) | |
| **ART adherence status** | | | | | |
| Adherent | Ref | 50.95 | <0.001 | Ref | **0.010** |
| Non-adherent | 2.87 (2.19-3.74) | | | 1.75 (1.14-2.67) | |

## Discussion

To our knowledge this is the first study that has estimated the prevalence of tobacco smoking; and SLT and determined the socio-demographic, clinical and behavioral characteristics among the PLWH receiving routine clinical care in Zambia. The study has shown that the prevalence of tobacco smoking and smokeless tobacco smoking is 9.7% and 1.4% respectively. Males smoked tobacco more than they used smokeless tobacco, contrary to women who used smokeless tobacco than tobacco smoking. Being male, having secondary level of education, alcohol consumption, being overweight/obese and non-adherence to ART were associated with the likelihood to smoking tobacco, while income of the patient, alcohol consumption and non-adherence to ART were associated with a higher likelihood to SLT.

The prevalence of the PLWH who smoke tobacco and receiving routine clinical care is 9.7% surprisingly, its lower than what is recorded in previous studies of the same settings [5,10,20]. Generally, literature has shown that PLWH use tobacco products two to three times more than the general population [19,21,22], this was our expectation, however, the prevalence of tobacco use among PLWH in our study did not differ from that of the general population, which is unexpected. One possible explanation is that our study may have captured a subset of PLWH who are more health-conscious or adherent to healthcare recommendations, resulting in a prevalence similar to that of the general population. The most plausible reason, however, could be underreporting due to social desirability bias. PLWH who actively seek healthcare may be more inclined to underreport tobacco use due to stigma or concerns about being advised to quit, potentially leading to a lower reported prevalence than actual usage rates. To minimize social desirability bias in future research on tobacco use among PLWH, utilizing digital or paper-based self-administered surveys instead of face-to-face interviews can reduce pressure on participants to provide socially desirable responses. Where resources permit, supplementing self-reported data with objective measures, such as biochemical verification (e.g., cotinine tests for tobacco use), can enhance accuracy and help identify potential discrepancies

When the population was categorized by gender, it was not surprising to note that our study showed a higher prevalence of tobacco smoking in men (21.9%) compared to women (3.7%). This conforms with previous HIV/tobacco studies. A recent HIV/tobacco study by Mdege et al. using the Demographic Health Survey (DHS) [4] showed that 24.4% and 1.3% of men and women smoked tobacco, respectively. Similarly, Kress et al [19] using DHS showed that 19.5% and 2.7% of men and women smoked tobacco, respectively. From these two recent studies, our study has revealed a concerning rise in the prevalence of tobacco smoking among women, marking a significant public health issue. Regarding smokeless tobacco, it's not surprising to see that the proportion of WLWH who use smokeless tobacco is higher than that of men. This conforms with previous studies. [23,24].

**Table 4. Factors associated with smokeless tobacco use among PLHIV.**

| Characteristics | Crude OR (95% CI) | chi2 Statistic | p-value | Adjusted OR (95% CI) | p-value |
|---|---|---|---|---|---|
| **Gender** | | | | | |
| Female | Ref | 2.40 | 0.121 | Ref | 0.287 |
| Male | 1.45 (0.91-2.30) | | | 0.70 (0.36-1.36) | |
| **Age (years)** | | | | | |
| 18-29 | Ref | 6.30 | 0.098 | Ref | 0.698 |
| 30-44 | 2.21 (0.93-5.24) | | | 1.71 (0.49-5.91) | |
| 45-59 | 1.65 (0.67-4.04) | | | 1.47 (0.40-5.42) | |
| 60+ | 0.86 (0.21-3.44) | | | 0.67 (0.07-6.86) | |
| **Education** | | | | | |
| No formal schooling | Ref | 7.71 | 0.052 | | |
| Primary | 0.77 (0.42-1.41) | | | | |
| Secondary | 0.63 (0.36-1.09) | | | | |
| College/University | 0.21 (0.05-0.88) | | | | |
| **Marital status** | | | | | |
| Never married | Ref | 0.83 | 0.660 | | |
| Currently married/Cohabitating | 1.07 (0.52-2.22) | | | | |
| Divorced/Separated/Widowed | 1.32 (0.61-2.83) | | | | |
| **Employment status** | | | | | |
| Formal employment | Ref | 1.13 | 0.567 | | |
| Informal employment | 0.72 (0.39-1.32) | | | | |
| Unemployed | 0.85 (0.47-1.53) | | | | |
| **Average yearly income** | | | | | |
| ≤4,000 ZMW | Ref | 9.46 | 0.002 | Ref | **0.007** |
| >4,000 ZMW | 0.35 (0.16-0.74) | | | 0.31 (0.14-0.73) | |
| **Location** | | | | | |
| Rural | Ref | 1.96 | 0.161 | | |
| Urban | 1.45 (0.85-2.47) | | | | |
| **Behavior variables** | | | | | |
| **Consumed alcohol within past 12m** | | | | | |
| No | Ref | 22.07 | <0.001 | Ref | **0.008** |
| Yes | 10.06 (2.45-41.31) | | | 14.74 (1.99-109.02) | |
| **Clinical variables** | | | | | |
| **BMI** | | | | | |
| Normal weight (18.5–24.9) | Ref | 6.94 | 0.074 | | |
| Underweight (≤18.4) | 1.19 (0.58-2.43) | | | | |
| Overweight (25–29.9) | 0.51 (0.26-1.00) | | | | |
| Obese (≥30) | 0.5 (0.20-1.26) | | | | |
| **Told by a doctor or other health worker that you have raised BP or HTN**\*\*** | | | | | |
| No | Ref | 2.77 | 0.096 | Ref | 0.122 |
| Yes | 1.62 (0.94-2.78) | | | 1.81 (0.85-3.85) | |

*(Continued)*

**Table 4.** (Continued)

| Characteristics | Crude OR (95% CI) | chi2 Statistic | p-value | Adjusted OR (95% CI) | p-value |
|---|---|---|---|---|---|
| **Told by a doctor or other health worker that you have raised blood sugar or diabetes** | | | | | |
| No | Ref | 2.71 | 0.100 | | |
| Yes | 2.45 (0.93-6.41) | | | | |
| **Heart attack or stroke** | | | | | |
| No | Ref | 0.43 | 0.514 | Ref | 0.851 |
| Yes | 1.67 (0.40-6.9) | | | 1.22 (0.16-9.46) | |
| **ART adherence status** | | | | | |
| Adherent | Ref | 8.84 | 0.003 | Ref | **0.002** |
| Non-adherent | 2.79 (1.52-5.13) | | | 3.32 (1.54-7.17) | |

After adjusting for socio – demographic and clinical factors, our analyses on both tobacco smoking and SLT model showed that men were significantly more likely to smoke tobacco than women; and women were more likely to use SLT than men although this could not reach statistical significancy. The proportion of those using SLT our study was low (1.4%), limiting our ability to make broad inferences about this segment of our population. Over the past two decades, research has highlighted the influence of gender-related factors on smoking behavior [17,25]. Pampel et al [26] examined population-based data from 16 DHS of men aged 15–54 years and women aged 15–49 years in 14 sub-Saharan countries and showed that although tobacco use varies widely across sub-Saharan African countries, one consistent pattern is that smoking rates are higher among men than women. Social norms and taboos play a significant role. Women face stronger cultural restrictions against smoking, which contributes to their lower prevalence. In contrast to SLT, women tend to use SLT more than men, this could be due to the myths and perceptions hence not surprising to see that woman living with HIV (WLWH) used SLT more than men in our study. Our findings are not different from other studies in the region [23]. What's of public health concern is that more than half of the WLWH in our study used SLT. The plausible explanation could be that its culturally acceptable mode of using tobacco in women. It's also could be driven mainly by these strong myths and perceptions that smokeless tobacco particularly Insunko enhances sexual enjoyment as it dries and makes the vaginal tight; and that it's an immunity booster and aid in viral suppression in PLWH.

Educational level, and income levels have been used as an individual's economic status, each one of them using a different causal pathway to explain the health outcomes [27]. Our study revealed a notable inverse relationship between education level and tobacco smoking. Smokeless tobacco though showed the same association, it was not statistically significant. As an individual's education level increases, the likelihood of tobacco smoking decreases. This finding aligns with prior research conducted in both developed and developing countries [28–30]. Education plays a crucial role in shaping health behavior decisions in that individuals with higher education tend to make more informed choices leading to better health outcomes. As regards to income, only the regression model for smokeless tobacco use showed a statistical significance. Those with lower income ≤4,000 ZMW were more likely to use SLT than those with >4,000 ZMW. A plethora of literature has shown that prevalence of smoking and tobacco use is highest in the lower income as compared the high income [30,31]. Insunko sold in sachets is cheap and affordable, women can easily access the product even within the neighborhood.

Numerous studies have demonstrated a link between alcohol consumption and tobacco smoking [32,33], it's not surprising that almost 75% of the tobacco smokers and 80% of SLT users reported alcohol consumption. Our study showed correlation between alcohol consumption and tobacco smoking; between alcohol consumption and SLT, with those who

consume alcohol being almost 5 times more likely to smoke tobacco while being almost 14 times more likely to use SLT. Several mechanisms may contribute to the strong association between alcohol consumption and tobacco use, Swan et al, propose that genetic factors contribute significantly to the covariance of both behaviors [34], while Niaura et al, advances the stress-coping theory that suggests that alcohol and tobacco serve as coping mechanisms to regulate emotions [35].

In conformity with other studies [36–38], our study also showed that there is a significant association between underweight/overweight/obesity and tobacco smoking. Tobacco smokers weigh less than the non-smokers because the nicotine affects weight through biological and behavior mechanisms [36,37,39]. The physiological effect of nicotine is that it results in appetite suppression, increase resting metabolism rates and reduced calorie storage [40–42]. Smoking is also related to decrease to meal size [40], and some of the smokers use smoking as a marker for meal termination in that instead of eating they would substitute smoking [42]. This association was not noted in PLWH who use smokeless tobacco. Our study has showed that the PWLH who smoke tobacco and use smokeless tobacco are less likely to adhere to ART. Previous studies have indicated that ART adherence in PLWH is a significant problem [43] but more so in PLWH who smoke [44,45]. Aggarwal et al. have reported that smokers are less adherent to other medications as well [46]. A potential explanation may be that PLWH who smoke tobacco and use smokeless tobacco are less likely to engage in self-care behaviors such as ART adherence compared with non-smokers. Furthermore, Zambia faces pressing public health challenges among PLWH, with tobacco use intensifying health risks amid the country's dual burden of infectious and non-communicable diseases. Despite significant progress in HIV management through widespread ART access, tobacco use remains an often-overlooked factor that threatens to undermine this advancement. The strong association between tobacco use and non-adherence to ART underscores the urgency of integrating tobacco cessation interventions within HIV care settings. Given Zambia's existing healthcare infrastructure, primary care facilities and ART clinics present a strategic opportunity to implement comprehensive tobacco cessation programs, ensuring a more holistic approach to patient care.

Our study is not without some limitations. Due to the observational design, we were unable to assess changes in these factors over time. This design allowed us to identify factors associated with tobacco smoking and smokeless tobacco use but did not permit us to establish causality. Although we adjusted for known confounders in the multivariable model, the potential for residual confounders inherent in observational studies remains, which might affect the interpretation of our findings. The data relied on self-reported responses, which may be subject to information bias, potentially leading to the misclassification of tobacco smoking and smokeless tobacco use. Due to study limitations, we were unable to utilize biochemical markers, such as cotinine tests, to validate these responses and detect discrepancies. Additionally, the generalizability of our findings is limited as this was a single cross-sectional study conducted in Zambia. The exclusion of the pregnant women and breastfeeding mothers from the study, while justified for ethical and health-related reasons, may have implications for the generalizability of the findings. Given that tobacco use can significantly impact maternal and infant health, understanding smoking behaviors among this subgroup of PLWH would have been valuable for shaping targeted interventions. Their exclusion means that the study may not fully capture the smoking patterns, cessation challenges, and health outcomes specific to this vulnerable population. To our knowledge, this is the first study in Zambia to determine national tobacco smoking and smokeless tobacco use among PLWH. It had a large sample size and was weighted based on the HIV burden at the district level. Surveyed PLWH were randomly selected from well-established ART clinics across the country, making the results generalizable to the general HIV population in Zambia.

In conclusion, our findings underscore the importance of embedding tobacco cessation strategies within Zambia's existing HIV care framework. Healthcare facilities offering ART services should integrate routine screening for tobacco use and provide counseling or referrals to cessation support programs. Strengthening collaborations between public health authorities and community organizations could ensure broader outreach, particularly to low-income groups more vulnerable to SLT use. Policy-wise, reinforcing tobacco control regulations and enhancing public awareness campaigns would be instrumental in reducing tobacco use among PLWH. Furthermore, addressing alcohol consumption—given its strong association with both smoking and SLT use through targeted behavioral interventions in HIV clinics could yield positive outcomes.

By contextualizing the findings within Zambia's healthcare challenges and aligning proposed interventions with existing resources, our study contributes to a more actionable framework for reducing tobacco use among PLWH in Zambia.

## Acknowledgments

We extend our deepest appreciation to all individuals and institutions whose support and contributions were critical to the successful implementation of this project. This endeavor would not have been possible without their invaluable assistance.

We thank the Ministry of Health and the Global Fund through the Prevention of HIV, TB, and Malaria for their financial and technical support. We are also profoundly grateful to the Ministry of Health Leadership, particularly the Non-Communicable Disease Unit, for their oversight and guidance throughout the study. Their expertise and support were essential in maintaining the highest standards of ethics and safety.

Additionally, we offer heartfelt thanks to the research assistants for their valuable contributions to data collection and analysis, which were fundamental to the project's success. We also acknowledge all the investigators for their crucial roles, providing expertise, guidance, and support pivotal to this study. We also extend our gratitude the project management team for their tireless efforts in overseeing daily activities, ensuring efficient planning and execution within the stipulated timeframe and budget. Finally, we express special thanks to all participants who agreed to take part in the study without them this project was not going to be successful.

## Author contributions

**Conceptualization:** Cosmas Zyambo, Chomba Mandyata, Henry Phiri, Malizgani P. Chavula, Hikabasa Halwindi, Joseph Zulu, Wilbroad Mutale.

**Data curation:** Cosmas Zyambo, Paul Somwe.

**Formal analysis:** Cosmas Zyambo, Paul Somwe.

**Funding acquisition:** Henry Phiri, Hikabasa Halwindi, Joseph Zulu.

**Methodology:** Cosmas Zyambo, Chomba Mandyata, Mwiche Musukuma, Phoebe Bwembya, Henry Phiri, Malizgani P. Chavula, Hikabasa Halwindi, Joseph Zulu, Wilbroad Mutale.

**Project administration:** Mwiche Musukuma, Henry Phiri, Malizgani P. Chavula, Hikabasa Halwindi, Joseph Zulu, Wilbroad Mutale.

**Supervision:** Cosmas Zyambo, Chomba Mandyata, Mwiche Musukuma, Hikabasa Halwindi, Joseph Zulu, Wilbroad Mutale.

**Validation:** Hikabasa Halwindi.

**Visualization:** Cosmas Zyambo, Hikabasa Halwindi, Wilbroad Mutale.

**Writing – original draft:** Cosmas Zyambo.

**Writing – review & editing:** Cosmas Zyambo, Paul Somwe, Chomba Mandyata, Mwiche Musukuma, Phoebe Bwembya, Henry Phiri, Malizgani P. Chavula, Hikabasa Halwindi, Joseph Zulu, Wilbroad Mutale.

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
