## [Decision Letter · Decision Letter 0]

Dear Dr. Zyambo,

Thank you for submitting your manuscript to PLOS ONE. After careful consideration, we feel that it has merit but does not fully meet PLOS ONE’s publication criteria as it currently stands. Therefore, we invite you to submit a revised version of the manuscript that addresses the points raised during the review process.

The manuscript has received an overall positive feedback from the reviewers. However, there seems some major recommendations on the clarity of the messages conveyed. You are requested to kindly revise as per the comments mentioned below. Please submit the versions with tracked changes turned on, along with an in-line reply to individual reviewers comments so that the changes may easily be tracked during the re-review process.

We look forward to receiving your revised manuscript.

Kind regards,

Yogesh Kumar Jain, MPH

Academic Editor

PLOS ONE

Journal Requirements:

Global Fund

4. In the online submission form, you indicated that the data is available if requested from the university of Zambia and the Ministry of Health in Zambia

Reviewers' comments:

Reviewer's Responses to Questions

**Comments to the Author**

1. Is the manuscript technically sound, and do the data support the conclusions?

Reviewer #1: Yes

Reviewer #2: Yes

Reviewer #3: Yes

2. Has the statistical analysis been performed appropriately and rigorously?

Reviewer #1: Yes

Reviewer #2: Yes

Reviewer #3: Yes

3. Have the authors made all data underlying the findings in their manuscript fully available?

Reviewer #1: Yes

Reviewer #2: No

Reviewer #3: Yes

4. Is the manuscript presented in an intelligible fashion and written in standard English?

Reviewer #1: No

Reviewer #2: No

Reviewer #3: Yes

Reviewer #1: Thank you for the opportunity to review this manuscript on tobacco use among people living with HIV (PLWH) in Zambia. The study addresses an important research gap regarding tobacco consumption patterns and associated factors in this population based on a nationwide cross-sectional survey.

Strengths

The study addresses a significant research gap in the region by examining tobacco use patterns among PLWH in Zambia

The nationwide cross-sectional design provides valuable population-level insights

The findings may have important implications for local health policy development

Major Concerns

Formatting and Clarity Issues

Multiple instances of formatting errors, punctuation mistakes, and incorrect abbreviation usage require thorough revision

Several ambiguous statements need clarification, particularly in the abstract. For example, it is unclear whether the gender-specific percentages (21.9% men and 3.7% women) are subsets of the 9.7% current tobacco smokers or represent different calculations

Statistical Methodology

The manuscript fails to specify how missing data for dependent variables was handled (full case analysis vs. interpolation)

There is no mention of strategies to address multiple comparisons when assessing associations of multiple factors with two outcome variables, potentially increasing Type I error risk

No evidence of multicollinearity assessment, which is crucial for ensuring model stability and accuracy

The absence of common goodness-of-fit indicators (e.g., pseudo-R², Hosmer-Lemeshow test) limits evaluation of model reliability and applicability

Contextual Integration

The discussion would benefit from better integration with the current state of Zambia's healthcare system

More consideration of how findings relate to local health infrastructure and policy would enhance practical relevance

Recommendations

Conduct a thorough review of the manuscript for grammatical errors, formatting inconsistencies, and unclear abbreviations

Clarify ambiguous statistical presentations, especially in the abstract

Provide complete information on statistical methodology, including missing data handling, multiple comparison adjustments, multicollinearity testing, and goodness-of-fit measures

Strengthen the discussion by better contextualizing findings within Zambia's healthcare landscape and considering implementation implications

For more specific comments, please refer to the annotations in the document.

Reviewer #2: I would like to congratulate the authors on this fine piece of work, exploring an important public health issue in SSA. I would recommend a major revision of the manuscript before acceptance.

In line 120, the authors mentioned that “Because the main focus of the parent study was NCD’s among PLWH, all pregnant women and breastfeeding mothers were excluded from the study”. Please kindly explain the rationale of including age group of 18-29 years old.

Sampling yielded a result of 67.2% females and 32.8% males PLWH (line 182). Is this generally representative of the statistics of PLWH living in the community among SSA? If this is not so, please explain why you think the study results deviate from statistics in the community. Please do input these thoughts under discussion.

In line 267, the authors mentioned that the unexpected finding was due to under reporting due to social desirability bias which could be the case. What is another possible explanation for this difference?

In lines 355-363, the authors should consider rephrasing the sentences, paying attention to sentence structuring and how future direction should be undertaken with respect to the study findings. Do take note that the associations are not causal as the authors have rightly pointed out. Henceforth, one should be careful in mentioning the intervention and associating them with subsequent reduction in tobacco use. For example, does improving financial means truly reduce reliance on smokeless tobacco use or is education an intermediary between financial situation and tobacco use?

Reviewer #3: -The introduction mentions that "Zambia currently lacks a structured smoking cessation policy for individuals attending routine HIV clinics". It might be strengthened by briefly discussing the potential impact of such a policy and how the study findings could inform its development.

-pregnant women and breastfeeding mothers are excluded from your study. While the reasoning is provided, it would be useful to briefly discuss the potential implications of this exclusion on the generalizability of the findings.

-Could you elaborate on the strategies that could be used to minimize social desirability bias in future research on tobacco use among PLWH?

-What specific clinical and public health interventions do you recommend based on your findings, and how could these be implemented in the Zambian context?

**Do you want your identity to be public for this peer review?** For information about this choice, including consent withdrawal, please see our Privacy Policy

Reviewer #1: No

Reviewer #2: No

Reviewer #3: **Yes: ** Adane Mekonnen Gebrewold

---

## [Author Response · Author response to Decision Letter 1]

28 May 2025

Dear PLOS ONE: Editors and Staff,

On behalf of all authors of “Tobacco smoking and Smokeless tobacco use among People Living with HIV in Zambia Findings from a 2023 National NCD/HIV Survey” we sincerely thank you for the opportunity to resubmit our manuscript for consideration. We would also like to thank the reviewers whose thorough evaluation of our work has given us the opportunity to make substantive improvements. Below is a point-by-point response to the concerns and comments of the reviewers. Whenever possible we have endeavored to integrate the recommendations from the initial review.

On the funding, we add this statement “The funders had no role in study design, data collection and analysis, decision to publish, or preparation of the manuscript."

Again, we are appreciative of this opportunity to resubmit our work to PLOS ONE. With the help of the academic editor and reviewers, we believe we are resubmitting an improved text.

Sincerely,

Cosmas Zyambo

Journal Requirements:

Response: This has been done

Response: This has been done

Global Fund

Response: This has been done

4. In the online submission form, you indicated that the data is available if requested from the university of Zambia and the Ministry of Health in Zambia

Response: This paper is a series of papers that are still being written. Analysis is currently on going for other papers. All the data after the analysis will be submitted to the National health Ethics authority website provided https://www.nhra.org.zm/. For further details regarding the dataset, kindly reach out to the Principal Investigator, Prof. Wilbroad Mutale. wmutale@yahoo.com

Reviewer #1:

Reviewer #1: Thank you for the opportunity to review this manuscript on tobacco use among people living with HIV (PLWH) in Zambia. The study addresses an important research gap regarding tobacco consumption patterns and associated factors in this population based on a nationwide cross-sectional survey.

Strengths

The study addresses a significant research gap in the region by examining tobacco use patterns among PLWH in Zambia. The nationwide cross-sectional design provides valuable population-level insights. The findings may have important implications for local health policy development

Response: We thank you for these comments on the significance of our manuscript and for the recommendations. We have endeavored to address the concerns which have resulted in substantive improvements to the manuscript, as detailed below:

Major Concerns

Formatting and Clarity Issues

Multiple instances of formatting errors, punctuation mistakes, and incorrect abbreviation usage require thorough revision

Several ambiguous statements need clarification, particularly in the abstract. For example, it is unclear whether the gender-specific percentages (21.9% men and 3.7% women) are subsets of the 9.7% current tobacco smokers or represent different calculations

Response: The abstract has been modified to enhance clarity and readability. The statement “9.7% were current tobacco smokers (21.9% men, 3.7% women) ….” entails that of the 9,7% who are current smokers, 21.7% are males and 3.7% are females. They are enclosed in brackets

Statistical Methodology

The manuscript fails to specify how missing data for dependent variables was handled (full case analysis vs. interpolation).

Response:

Thank you for this important observation. We confirm that a complete case analysis was conducted on 1,356 observations. The reduction in sample size primarily resulted from a prerequisite question on whether a study participant had ever been tested for diabetes. Approximately more than half of the participants responded "No," and as such, the key variable indicating whether they had been told they had raised blood sugar or diabetes, which was included in the model, only had responses from those who answered "Yes" to being tested.

Furthermore, likelihood ratio tests used to determine the best-fitting model require consistent observations across all models being compared. This necessitated the exclusion of records with missing data on any of the model variables. Therefore, only complete cases were retained, and no imputation or interpolation methods were applied.

As shown in the screenshot below, complete observations were identified using a binary variable complete, where only observations with complete == 1 were retained for the multivariable analysis. This ensured that all included cases had non-missing values across all variables used in the models.

There is no mention of strategies to address multiple comparisons when assessing associations of multiple factors with two outcome variables, potentially increasing Type I error risk

Response:

We analysed the two outcome variables separately and did not apply formal adjustments for multiple comparisons.

No evidence of multicollinearity assessment, which is crucial for ensuring model stability and accuracy

Response:

We did conduct a multicollinearity assessment for all variables included in the model. As shown in the collinearity results image below, Spearman’s rho estimates indicate that none of the variables were strongly correlated.

The absence of common goodness-of-fit indicators (e.g., pseudo-R², Hosmer-Lemeshow test) limits evaluation of model reliability and applicability

Response: We did conduct the McFadden’s pseudo-R² value of 0.1921, which indicates a moderate model fit.

Contextual Integration

The discussion would benefit from better integration with the current state of Zambia's healthcare system

More consideration of how findings relate to local health infrastructure and policy would enhance practical relevance

Response: This has been added in the discussion as highlighted yellow in the manuscript “Zambia faces pressing public health challenges among PLWH, with tobacco use intensifying health risks amid the country’s dual burden of infectious and non-communicable diseases. Despite significant progress in HIV management through widespread ART access, tobacco use remains an often-overlooked factor that threatens to undermine this advancement. The strong association between tobacco use and non-adherence to ART underscores the urgency of integrating tobacco cessation interventions within HIV care settings. Given Zambia’s existing healthcare infrastructure, primary care facilities and ART clinics present a strategic opportunity to implement comprehensive tobacco cessation programs, ensuring a more holistic approach to patient care”

Response: The implementation implication of the study has been added in the conclusion as highlighted yellow in the manuscript” our findings underscore the importance of embedding tobacco cessation strategies within Zambia's existing HIV care framework. Healthcare facilities offering ART services should integrate routine screening for tobacco use and provide counseling or referrals to cessation support programs. Strengthening collaborations between public health authorities and community organizations could ensure broader outreach, particularly to low-income groups more vulnerable to SLT use. Policy-wise, reinforcing tobacco control regulations and enhancing public awareness campaigns would be instrumental in reducing tobacco use among PLWH. Furthermore, addressing alcohol consumption—given its strong association with both smoking and SLT use—through targeted behavioral interventions in HIV clinics could yield positive outcomes. By contextualizing the findings within Zambia's healthcare challenges and aligning proposed interventions with existing resources, our study contributes to a more actionable framework for reducing tobacco use among PLWH in Zambia”

Recommendations

Conduct a thorough review of the manuscript for grammatical errors, formatting inconsistencies, and unclear abbreviations

Clarify ambiguous statistical presentations, especially in the abstract

Provide complete information on statistical methodology, including missing data handling, multiple comparison adjustments, multicollinearity testing, and goodness-of-fit measures

Strengthen the discussion by better contextualizing findings within Zambia's healthcare landscape and considering implementation implications

Response: Thank you for the recommendation, these have been addressed as above and as highlighted yellow in the manuscript

Reviewer #2: I would like to congratulate the authors on this fine piece of work, exploring an important public health issue in SSA. I would recommend a major revision of the manuscript before acceptance.

Response: We thank you for these comments on the impotence of our manuscript and for the suggestions. We have endeavored to address the concerns which have resulted in substantive improvements to the manuscript, as detailed below:

In line 120, the authors mentioned that “Because the main focus of the parent study was NCD’s among PLWH, all pregnant women and breastfeeding mothers were excluded from the study”. Please kindly explain the rationale of including age group of 18-29 years old.

Response: The statement on the reason why the 18-29 years group was added has been added in the manuscript as highlighted in yellow. “Inclusion of the age group of 18–29 years in this study is crucial because young adults are at a pivotal life stage marked by transitions in education, employment, and social environments, all of which influence health behaviors, including tobacco use. Peer pressure, growing independence, and exposure to various stressors make them particularly vulnerable to initiating and sustaining tobacco use. From a healthcare perspective, understanding tobacco use patterns in this group enables early intervention, helping to mitigate long-term health risks, especially among PLWH, where tobacco exacerbates complications. Insights from this analysis can inform youth-focused prevention strategies, awareness campaigns, and policy measures designed to curb tobacco use before it becomes deeply ingrained.”

Sampling yielded a result of 67.2% females and 32.8% males PLWH (line 182). Is this generally representative of the statistics of PLWH living in the community among SSA? If this is not so, please explain why you think the study results deviate from statistics in the community. Please do input these thoughts under discussion.

Response: In Sub-Saharan Africa (SSA), women generally make up a higher proportion of people living with HIV (PLWH) compared to men. This is largely due to biological, social, and economic factors that increase women's vulnerability to HIV infection. While exact percentages vary by country, most studies indicate that women account for a significant majority of PLWH in the region. Our study’s finding of 67.2% females and 32.8% males aligns with broader trends observed in SSA, where women often represent over 60% of the HIV-positive population. Our study does not depart from literature in the SSA.

In line 267, the authors mentioned that the unexpected finding was due to under reporting due to social desirability bias which could be the case. What is another possible explanation for this difference?

Response: The other possible reason has been added in the manuscript as highlighted in yellow: “…. however, the prevalence of tobacco use among PLWH in our study did not differ from that of the general population, which is unexpected. One possible explanation is that our study may have captured a subset of PLWH who are more health-conscious or adherent to healthcare recommendations, resulting in a prevalence similar to that of the general population. The most plausible reason, however, could be underreporting due to social desirability bias. PLWH who actively seek healthcare may be more inclined to underreport tobacco use due to stigma or concerns about being advised to quit, potentially leading to a lower reported prevalence than actual usage rates”

In lines 355-363, the authors should consider rephrasing the sentences, paying attention to sentence structuring and how future direction should be undertaken with respect to the study findings. Do take note that the associations are not causal as the authors have rightly pointed out. Henceforth, one should be careful in mentioning the intervention and associating them with subsequent reduction in tobacco use. For example, does improving financial means truly reduce reliance on smokeless tobacco use or is education an intermediary between financial situation and tobacco use?

Response: we have restructured and modified this part for clarity and readability. The addition has been highlighted yellow in the manuscript. “….our findings underscore the importance of embedding tobacco cessation strategies within Zambia's existing HIV care framework. Healthcare facilities offering ART services should integrate routine screening for tobacco use and provide counseling or referrals to cessation support programs. Strengthening collaborations between public health authorities and community organizations could ensure broader outreach, particularly to low-income groups more vulnerable to SLT use. Policy-wise, reinforcing tobacco control regulations and enhancing public awareness campaigns would be instrumental in reducing tobacco use among PLWH. Furthermore, addressing alcohol consumption—given its strong association with both smoking and SLT use—through targeted behavioral interventions in HIV clinics could yield positive outcomes. By contextualizing the findings within Zambia's healthcare challenges and aligning proposed interventions with existing resources, our study contributes to a more actionable framework for reducing tobacco use among PLWH in Zambia”

Reviewer #3: -The introduction mentions that "Zambia currently lacks a structured smoking cessation policy for individuals attending routine HIV clinics". It might be strengthened by briefly discussing the potential impact of such a policy and how the study findings could inform its development.

Response: This sentence has been added (as highlighted in yellow) to strengthen the introduction on the cessation policy and how the study might add to the cessation policy formulation.” A structured smoking cessation policy could significantly

---

## [Decision Letter · Decision Letter 1]

Tobacco smoking and Smokeless tobacco use among People Living with HIV in Zambia Findings from a 2023 National NCD/HIV Survey

PONE-D-25-05511R1

Dear Dr. Zyambo,

We’re pleased to inform you that your manuscript has been judged scientifically suitable for publication and will be formally accepted for publication once it meets all outstanding technical requirements.

Kind regards,

Yogesh Kumar Jain, MPH

Academic Editor

PLOS ONE

Additional Editor Comments (optional):

Reviewers' comments:

Reviewer's Responses to Questions

**Comments to the Author**

Reviewer #1: All comments have been addressed

2. Is the manuscript technically sound, and do the data support the conclusions?

Reviewer #1: Yes

3. Has the statistical analysis been performed appropriately and rigorously?

Reviewer #1: Yes

4. Have the authors made all data underlying the findings in their manuscript fully available?

Reviewer #1: Yes

5. Is the manuscript presented in an intelligible fashion and written in standard English?

Reviewer #1: Yes

Reviewer #1: I am of the opinion that the quality of the paper has been significantly improved after this revision and there are no longer any significant revisions. The paper is innovative to a certain extent, the research content is abundant, the methodology is scientific and reasonable, the results are reliable, and it has certain reference value for the research in related fields, so it is suggested that it can be considered for acceptance for publication. However, it is hoped that the authors will again scrutinize the paper for typos, improper use of punctuation and other minor problems during the subsequent proofreading process to ensure the perfect presentation of the paper.

**Do you want your identity to be public for this peer review?** For information about this choice, including consent withdrawal, please see our Privacy Policy

Reviewer #1: No

---

## [Editor Report · Acceptance letter]

PONE-D-25-05511R1

PLOS ONE

Dear Dr. Zyambo,

I'm pleased to inform you that your manuscript has been deemed suitable for publication in PLOS ONE. Congratulations! Your manuscript is now being handed over to our production team.

Kind regards,

on behalf of

Dr. Yogesh Kumar Jain

Academic Editor

PLOS ONE